# Comparison of Percutaneous Coronary Intervention and Coronary Artery Bypass Grafting in NSTEMI Patients with Chronic Kidney Disease

**DOI:** 10.3390/medicina62010036

**Published:** 2025-12-24

**Authors:** Ali Palice, Ömer Faruk Çiçek, Ayşe Emre

**Affiliations:** 1Akçakale State Hospital, Akçakale 63500, Turkey; 2Department of Cardiology, Mehmet Akif Inan Education and Research Hospital, Sanliurfa 63300, Turkey; 3Department of Cardiology, Siyami Ersek Thoracic and Cardiovascular Surgery Center, Istanbul 34668, Turkey

**Keywords:** coronary artery bypass grafting, chronic kidney disease, myocardial infarction, percutaneous coronary intervention, SYNTAX score

## Abstract

*Background and Objectives:* Chronic kidney disease (CKD) is frequently observed among patients with non–ST elevation myocardial infarction (NSTEMI) and is associated with increased morbidity and mortality. Evidence comparing long-term outcomes after percutaneous coronary intervention (PCI) and coronary artery bypass grafting (CABG) in this high-risk population remains limited. The objective was to compare long-term major adverse cardiac event (MACE) outcomes between PCI and CABG in NSTEMI patients with CKD and multivessel disease. *Materials and Methods:* A total of 150 consecutive NSTEMI patients with CKD who underwent PCI or CABG were included in this retrospective observational cohort study. Patients were classified as having mild or moderate-to-severe CKD based on eGFR. Long-term outcomes included MACE (death, myocardial infarction, or ischemia-driven revascularization). Kaplan–Meier analysis was used to compare long-term MACE-free survival between groups. *Results:* PCI (*n* = 68) and CABG (*n* = 82) groups demonstrated comparable long-term MACE-free survival (log-rank *p* = 0.41). One-year MACE-free survival rates were 78% and 82%, respectively. Ischemia-driven revascularization was more frequent after PCI (*p* = 0.028), whereas major bleeding occurred more commonly after CABG (*p* = 0.003). *Conclusions:* In NSTEMI patients with CKD and multivessel disease, PCI and CABG provide comparable long-term MACE-free survival. Despite higher rates of repeat revascularization after PCI and greater bleeding risk after CABG, overall long-term outcomes were similar. CKD severity did not significantly modify treatment-related differences.

## 1. Introduction

Chronic kidney disease (CKD) affects approximately 40% of patients presenting with acute coronary syndrome (ACS) and is linked to elevated morbidity and mortality [1,2]. Despite the high-risk profile of these patients, guideline-recommended therapies and revascularization strategies are used less frequently due to concerns regarding peri-procedural complications and worsening renal function [3].

Patients with NSTEMI and CKD are significantly less likely to undergo invasive evaluation or percutaneous coronary intervention (PCI), even though PCI may reduce in-hospital mortality [4]. However, the management of coronary disease in CKD remains challenging. The ISCHEMIA-CKD trial demonstrated no mortality benefit of a routine invasive strategy in stable CKD patients, but these findings cannot be generalized to NSTEMI because ACS patients were largely excluded [5]. Current guidelines recommend an individualized revascularization strategy in NSTEMI patients with multivessel disease, balancing ischemic risk, anatomical complexity, and procedural risk. Patients with NSTEMI and CKD represent a particularly vulnerable population due to their high ischemic burden, increased bleeding risk, and frequent exclusion from randomized trials [6].

Revascularization decisions are further complicated by coronary anatomical complexity. Tools such as the SYNTAX score II 2020 provide individualized outcome estimates and may help guide the selection between PCI and coronary artery bypass grafting (CABG) in multivessel disease [6]. However, direct comparisons of PCI and CABG specifically in NSTEMI patients with CKD remain scarce.

Patients presenting with NSTEMI in the setting of multivessel coronary artery disease and chronic kidney disease constitute a particularly high-risk and heterogeneous population that is underrepresented in randomized revascularization trials. Thus, this study aimed to evaluate long-term clinical outcomes of PCI versus CABG in NSTEMI patients with CKD and multivessel coronary artery disease, and to assess the influence of CKD severity on treatment-related prognosis.

## 2. Methods

This study was designed as a retrospective observational cohort study including 150 consecutive patients diagnosed with non-ST-elevation myocardial infarction (NSTEMI) and chronic kidney disease (CKD) who underwent coronary revascularization—either percutaneous coronary intervention (PCI) or coronary artery bypass grafting (CABG)—for multivessel coronary artery disease at our institution between January 2017 and December 2018. Multivessel disease was defined as ≥70% stenosis in at least two major epicardial coronary arteries.

All procedures were conducted in accordance with the ethical standards of the institutional review board and the Declaration of Helsinki. Institutional ethics approval was obtained in 2019 (protocol number: 488665165-302.14.01). Because of the retrospective design, written informed consent was obtained from all patients at the time of their initial hospitalization.

Baseline demographic, clinical, laboratory, and treatment-related data were retrieved from hospital records. NSTEMI was defined according to the current European Society of Cardiology (ESC) guidelines for acute coronary syndrome without persistent ST-segment elevation [7]. CKD was classified based on the estimated glomerular filtration rate (eGFR), calculated using the Modification of Diet in Renal Disease (MDRD) equation. CKD was defined as eGFR < 60 mL/min/1.73 m^2^ and categorized as mild (eGFR 45–60) or moderate-to-severe (eGFR < 45) [8].

Patients were grouped according to the revascularization strategy (PCI vs. CABG). Revascularization strategy was determined by a multidisciplinary heart team based on clinical presentation, coronary anatomy, comorbidities, and patient preference. In the PCI group, all interventions were performed using Resolute Integrity drug-eluting stents. In the CABG group, the left internal mammary artery (LIMA) was grafted to the left anterior descending artery (LAD), and saphenous vein grafts were used for other target vessels. Venous blood samples were analyzed within 60 min, and complete blood count was obtained using an automated hematology analyzer (MIND-RAY BC-6800, Shenzhen, China). Left ventricular ejection fraction (LVEF) was measured by transthoracic echocardiography at admission.

The Global Registry of Acute Coronary Events (GRACE) risk score was calculated for each patient and categorized as <109, 109–140, or >140. The mean follow-up period was 517 days. Major adverse cardiac events (MACE) were defined as all-cause mortality, nonfatal myocardial infarction, or ischemia-driven revascularization. Mortality was classified as all-cause mortality. In-hospital outcomes included acute renal failure, major bleeding, reinfarction, acute heart failure, length of hospital stay, and in-hospital mortality. In addition, a prespecified in-hospital composite endpoint was defined as acute renal failure and/or major bleeding. Myocardial infarction was adjudicated according to the Universal Definition of Myocardial Infarction. Ischemia-driven revascularization was defined as repeat coronary intervention performed in the presence of recurrent symptoms and/or objective evidence of myocardial ischemia, rather than routine angiographic findings. Major bleeding was defined as clinically overt bleeding requiring transfusion or surgical/interventional treatment during hospitalization. Acute renal failure was defined according to KDIGO criteria as an increase in serum creatinine ≥0.3 mg/dL within 48 h or ≥1.5 times baseline during hospitalization, or the need for new-onset renal replacement therapy.

Exclusion criteria included end-stage renal disease requiring hemodialysis, prior renal transplantation, age <18 years, previous CABG or valve surgery, bare-metal stent implantation, mitral valve disease requiring surgical intervention, significant left main coronary stenosis (>50%), liver disease, active malignancy, prior blood transfusion, hematologic disorders, acute or chronic infection, autoimmune disease, and use of immunosuppressive therapy. Patients with significant left main coronary artery stenosis (>50%) were excluded because left main disease represents a distinct anatomical and prognostic entity with guideline-directed revascularization strategies, which could confound comparisons between PCI and CABG in non–left main multivessel disease.

### Statistical Analysis

An a priori power analysis was performed using G*Power version 3.1 to determine the minimum required sample size. Based on expected differences in major adverse cardiac events (MACE) between revascularization strategies, a two-sided comparison of proportions with α = 0.05, 1–β = 0.80, and an effect size (w) of 0.30 indicated that at least 144 patients were required to achieve adequate statistical power. The final analytical sample of 150 patients exceeded this threshold.

Statistical analyses were conducted using SPSS version 22.0 (SPSS Inc., Chicago, IL, USA) and Jamovi version 2.2.5. Continuous variables were presented as mean ± standard deviation or median (minimum–maximum), depending on distribution. Categorical variables were summarized as frequencies and percentages. The normality of the distribution was assessed using the Shapiro–Wilk, Kolmogorov–Smirnov, and Anderson–Darling tests. Missing data were minimal and were handled using complete-case analysis. No imputation methods were applied. Variables with missing values were excluded from the relevant analyses.

Between-group comparisons (PCI vs. CABG) were performed using the independent samples *t*-test for normally distributed continuous variables and the Mann–Whitney U test for non-normally distributed variables. Categorical variables were compared using the Pearson Chi-square test or Fisher’s Exact test where appropriate; for variables with more than two categorical levels, the Fisher–Freeman–Halton test was applied. These procedures correspond to the analyses presented in Table 1, Table 2, Table 3, Table 4 and Table 5.

The relationships between GRACE and SYNTAX scores within each revascularization group were evaluated using Spearman’s correlation coefficients. Long-term event-free survival was analyzed using Kaplan–Meier curves, and differences between groups were assessed with the log-rank test (Figure 1).

Univariate Cox proportional hazards regression analysis was used to identify potential predictors of major adverse cardiac events (MACE). Variables with *p* < 0.10 in univariate analyses were included in the multivariate Cox regression model. Hazard ratios (HR) with 95% confidence intervals (CI) were reported. A two-sided *p*-value < 0.05 was considered statistically significant.

## 3. Results

A total of 150 patients (mean age 67.2 ± 8.4 years, 65.3% male) with NSTEMI and chronic kidney disease (CKD) who underwent coronary revascularization were included. The median follow-up duration was 517 days (minimum 2 days, maximum 1341 days). Of these, 68 (45.3%) received PCI and 82 (54.7%) underwent CABG. Age and sex distributions were similar between groups (*p* = 0.069 and *p* = 0.313). Median BMI was significantly lower in the CABG group compared with the PCI group (*p* = 0.049), although the prevalence of obesity did not differ (*p* = 1.000). Hypertension (81.3%), diabetes mellitus (62.7%), and hyperlipidemia (48.7%) were the most common comorbidities. Active smoking was more frequent in the CABG group (*p* = 0.029). Coronary artery disease and hyperlipidemia were more common in the PCI group (*p* < 0.001 and *p* = 0.001), whereas peripheral artery disease and COPD were more frequent in the CABG group (*p* = 0.038 and *p* = 0.037). Other baseline features are presented in Table 1.

Hemodynamic and echocardiographic parameters are shown in Table 2. Heart rate and diastolic blood pressure differed significantly between groups (*p* = 0.001 and *p* = 0.030). Killip class distribution was similar (*p* = 0.298). The GRACE score was significantly higher in the PCI group (*p* = 0.004). High-risk GRACE categories (>140) were also more common in the PCI (*p* = 0.034). SYNTAX scores were significantly higher in the CABG group (*p* < 0.001), with more patients in the ≥33 category (*p* < 0.001) (Table 3).

In-hospital outcomes are shown in Table 4. The high rate of the prespecified in-hospital composite endpoint (acute renal failure and/or major bleeding) was mainly driven by acute renal failure and major bleeding events, while in-hospital mortality and reinfarction rates remained relatively low. Major bleeding occurred more frequently in the CABG group (20.7% vs. 2.9%, *p* = 0.003). Acute renal failure was more common in the PCI group, although this difference was not statistically significant (*p* = 0.071). Length of hospital stay was significantly longer in the CABG group (*p* < 0.001).

Long-term outcomes are presented in Table 5. Major adverse cardiac events occurred in 30.9% of the PCI group and 22.0% of the CABG group (*p* = 0.292). Ischemia-driven revascularization was significantly more frequent in PCI (19.1% vs. 6.1%, *p* = 0.028). Long-term mortality and nonfatal MI rates were similar between groups (*p* > 0.05).

In the CABG group, smoking (*p* = 0.003) and hyperlipidemia (*p* = 0.038) were associated with MACE in univariate analysis. In subgroup analyses, smoking status showed a statistically significant association with MACE in the CABG group; however, this finding should be interpreted with caution and was not suggestive of a protective effect. In Cox regression analyses, the revascularization strategy was not significantly associated with long-term major adverse cardiac events (MACE) in the overall cohort. Similar findings were observed in predefined subgroups stratified by renal function (eGFR < 45 mL/min/1.73 m^2^ and eGFR 45–60 mL/min/1.73 m^2^), both in unadjusted and adjusted models. The hazard ratios consistently crossed unity, indicating no statistically significant difference in long-term MACE risk between PCI and CABG during follow-up In multivariable Cox regression analysis, adjusted for age, sex, comorbidities, renal function, and Killip class, revascularization strategy (PCI vs. CABG) was not independently associated with long-term MACE, with hazard ratios crossing unity (Figure 1).

Kaplan–Meier analysis demonstrated no significant difference in long-term MACE-free survival between PCI and CABG (log-rank *p* = 0.41). Estimated survival curves did not diverge over time, indicating comparable long-term outcomes for both strategies (Figure 2).

## 4. Discussion

In this retrospective cohort of NSTEMI patients with chronic kidney disease and multivessel coronary artery disease, no statistically significant difference in long-term MACE-free survival was observed between PCI and CABG, despite distinct baseline clinical and anatomical risk profiles. Data on PCI and CABG in patients with chronic kidney disease (CKD) and multivessel coronary artery disease remain limited, largely because this high-risk population is frequently excluded or underrepresented in randomized controlled trials. For example, in the SYNTAX trial, only 264 patients had an eGFR < 60 mL/min/1.73 m^2^, underscoring the scarcity of evidence in CKD populations [9]. Although CABG was associated with fewer ischemic events than PCI in the CKD subgroup of SYNTAX, this difference was primarily driven by higher rates of ischemia-driven repeat revascularization after PCI, while mortality did not differ significantly between strategies [9]. Our findings are particularly relevant because they reflect a real-world NSTEMI population with multivessel disease and varying CKD severity, a combination that has rarely been evaluated together in prior studies.

Previous work has also demonstrated that CKD modifies outcomes differently following PCI and CABG. In a major study evaluating myocardial revascularization outcomes in diabetic patients with CKD, CABG was associated with a higher early risk but improved longer-term event reduction compared with PCI, highlighting that renal dysfunction significantly influences the comparative performance of revascularization strategies [10]. This supports the notion that CKD is a major determinant of procedural risk and must be integrated into treatment decision-making.

Advances in drug-eluting stent technology have reduced restenosis rates in CKD patients, especially compared with older bare-metal stents [11]. However, observational data suggest that CABG may be associated with higher early mortality and increased risk of progression to end-stage renal disease in the early postoperative period, possibly due to perioperative hemodynamic stress [12,13]. Despite this early risk, the overall cardiovascular mortality in CKD patients remains markedly elevated—approximately 5–10 times higher than the risk of progressing to dialysis—highlighting the importance of timely and effective coronary revascularization [12,13,14].

In our study, there were substantial baseline differences between the PCI and CABG groups in terms of anatomical and clinical risk profiles, with higher SYNTAX scores observed in the CABG group and higher GRACE scores in the PCI group. While these imbalances reflect real-world patient selection, they also indicate the presence of potential confounding factors that may have influenced long-term outcomes. Consistent with previous studies, our findings showed that long-term major adverse cardiac event (MACE) rates were comparable between PCI and CABG despite baseline differences in risk profiles. Earlier analyses have shown that PCI tends to offer lower short-term morbidity, whereas CABG provides more durable protection from ischemia-driven revascularization. This pattern was also observed in our cohort, where repeat revascularization was significantly more frequent after PCI, although mortality and myocardial infarction rates did not differ between groups [15,16]. Consistent with previous evidence, patients with chronic kidney disease undergoing CABG experience higher bleeding rates and longer hospital stays compared with PCI. This is largely attributed to uremia-related platelet dysfunction, perioperative anticoagulation, and increased susceptibility to postoperative complications in the CKD population [10,15].

The presence of heavily calcified coronary lesions in CKD may impair optimal stent expansion and drug delivery, contributing to higher restenosis and revascularization rates [17,18]. In our study, despite CABG patients having more complex anatomy—including higher SYNTAX scores and a greater proportion with SYNTAX ≥ 33—long-term outcomes remained similar to those of PCI patients. This suggests that surgical revascularization may mitigate the adverse impact of complex coronary disease in CKD, although definitive conclusions cannot be drawn without randomized data.

Importantly, we evaluated mild and moderate-to-severe CKD subgroups separately and found no significant differences in long-term outcomes between PCI and CABG across CKD severity levels, consistent with prior evidence showing comparable prognostic trends in advanced CKD patients [15]. This aligns with recent recommendations emphasizing that revascularization strategy in CKD should incorporate clinical presentation, anatomical complexity, and overall procedural risk rather than CKD stage alone [6,13]. In our study, the absence of a significant difference in outcomes across CKD severity subgroups should be interpreted with caution and does not imply equivalence. This finding may be related to the limited sample size, exclusion of dialysis-dependent patients, and classification of renal function based on a single baseline eGFR measurement.

Similarly, contemporary evidence emphasizes that in patients with multivessel coronary disease, the decision between PCI and CABG should be individualized by considering anatomical complexity and overall clinical risk rather than relying solely on renal function, supporting a tailored strategy in this population [19].

This study has several limitations. Baseline heterogeneity between the PCI and CABG groups—such as differences in SYNTAX score, GRACE score, comorbidity burden, smoking status, and chronic obstructive pulmonary disease—may have introduced selection bias and residual confounding despite statistical adjustment. First, it was a single-center observational study, which may further limit generalizability. Second, subgroup analyses according to CKD severity included relatively small numbers of patients. Third, renal function was assessed using a single baseline estimated glomerular filtration rate (eGFR) measurement, which may not fully capture chronic renal status. Fourth, coronary anatomy was evaluated using the SYNTAX score without complementary intravascular imaging (IVUS/OCT) or assessment of completeness of revascularization. In addition, contrast volume data during PCI and formal perioperative risk stratification scores (such as STS or EuroSCORE II) for CABG patients were not systematically available and therefore could not be incorporated into the analyses. Finally, although follow-up exceeded one year, longer observation periods are needed to evaluate very late adverse events.

## 5. Conclusions

In conclusion, no statistically significant difference in long-term MACE-free survival was observed between PCI and CABG in NSTEMI patients with chronic kidney disease and multivessel coronary artery disease. This finding was consistent across both mild and moderate-to-severe CKD subgroups. These results suggest that revascularization strategy may be guided by clinical presentation, anatomical complexity, and overall risk profile rather than CKD stage alone. Larger, multicenter studies with longer follow-up are needed to validate these findings and further refine revascularization decision-making in this high-risk population.

## Figures and Tables

**Figure 1 medicina-62-00036-f001:**
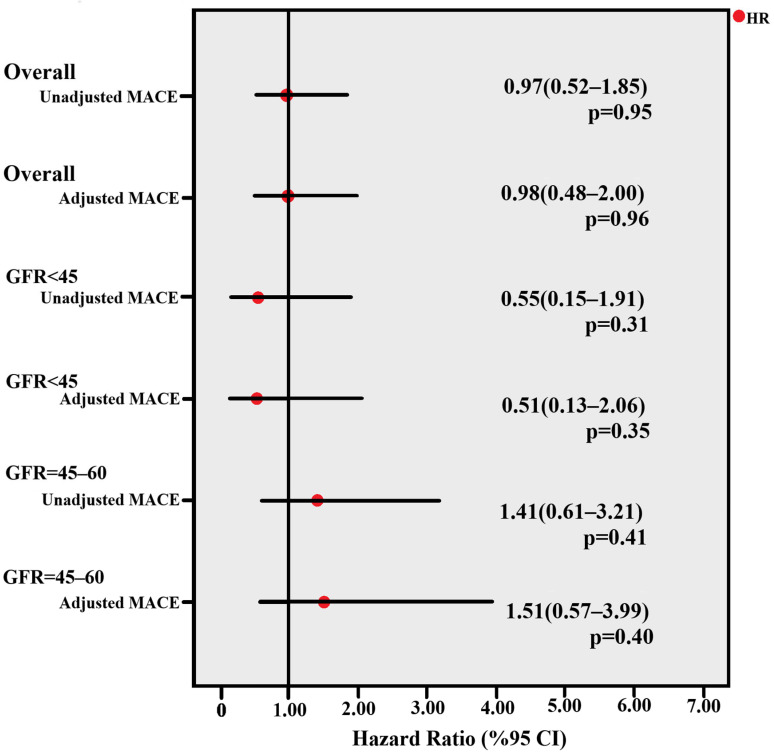
Cox regression analysis of the association between revascularization strategy and long-term major adverse cardiac events (MACE).

**Figure 2 medicina-62-00036-f002:**
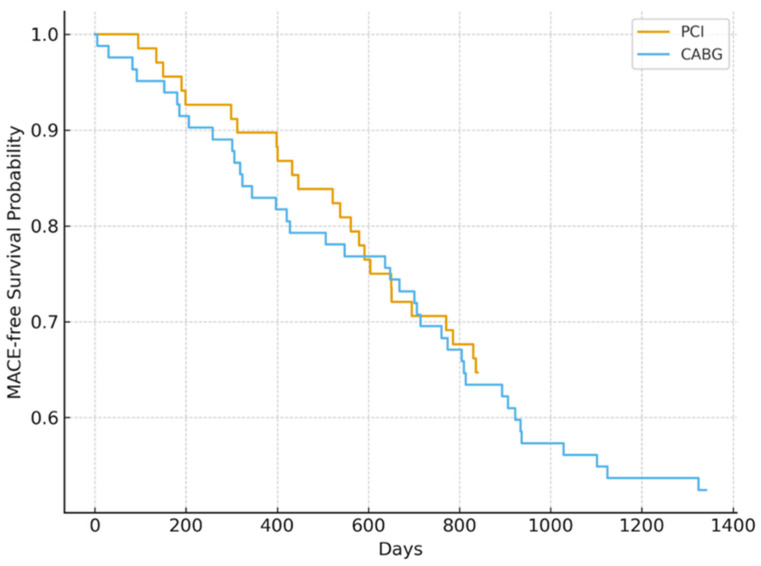
Kaplan–Meier curves for long-term MACE-free survival in NSTEMI patients with chronic kidney failure undergoing PCI or CABG.

**Table 1 medicina-62-00036-t001:** Demographic and clinical characteristics of the patients.

		Overall (*n* = 150)	Group PCI (*n* = 68)	Group CABG (*n* = 82)	*p*
**Age (year)** ^†^		67.2 ± 8.4	68.6 ± 8.1	66.1 ± 8.5	0.069
**Sex** ^‡^					
	Female	52 (34.7)	27 (39.7)	25 (30.5)	0.313
	Male	98 (65.3)	41 (60.3)	57 (69.5)	
**Body mass index (kg/m^2^)** ^§^		28.0 [18.0–46.0]	29.0 [23.0–46.0]	27.0 [18.0–40.0]	**0.049**
**Body mass index ≥ 30 kg/m^2^** ^‡^		63 (42.0)	29 (42.6)	34 (41.5)	1.000
**Smoking status** ^‡^	Smoking history	91 (60.7)	36 (52.9)	55 (67.1)	0.110
	Current smoker	55 (36.7)	18 (26.5)	37 (45.1)	**0.029**
	Non-smoker	55 (36.7)	32 (47.1)	23 (28.0)	0.025
**Comorbidities** ^‡^					
	Hypertension	122 (81.3)	56 (82.4)	66 (80.5)	0.935
	Diabetes mellitus	94 (62.7)	46 (67.6)	48 (58.5)	0.328
	Hyperlipidemia	73 (48.7)	44 (64.7)	29 (35.4)	**0.001**
	Coronary artery disease	67 (44.7)	42 (61.8)	25 (30.5)	**<0.001**
	Chronic obstructive pulmonary disease	30 (20.0)	8 (11.8)	22 (26.8)	**0.037**
	Peripheral artery disease	22 (14.7)	5 (7.4)	17 (20.7)	**0.038**
	Stroke	7 (4.7)	2 (2.9)	5 (6.1)	0.457
	eGFR, mL/dk/1.73 m^2^				
	<45 mL/dk/1.73 m^2^	50(33.3)	25 (36.8)	25 (30.5)	0.52
	45–60 mL/dk/1.73 m^2^	100(66.7)	43 (63.2)	57 (69.5)	0.42
**Chest pain** ^‡^		147 (98.0)	66 (97.1)	81 (98.8)	0.590
**Resuscitation history** ^‡^		2 (1.3)	1 (1.5)	1 (1.2)	0.999
**Family members with myocardial infarction** ^‡^		36 (24.0)	11 (16.2)	25 (30.5)	0.064

^‡^: *n* (%), ^†^: mean ± standard deviation, ^§^: median [min–max]; PCI: percutaneous coronary intervention, CABG: coronary artery bypass grafting.

**Table 2 medicina-62-00036-t002:** Hemodynamic, electrocardiographic, and echocardiographic findings of the patients in Groups PCI and CABG.

	Group PCI (*n* = 68)	Group CABG (*n* = 82)	*p*
**Heart rate (beat/minute)** ^§^	80.0 [51.0–130.0]	88.0 [55.0–124.0]	**0.001**
**Systolic blood pressure (mmHg)** ^†^	137.1 ± 19.3	136.1 ± 23.7	0.787
**Diastolic blood pressure (mmHg)** ^†^	77.0 ± 12.1	72.2 ± 15.0	**0.030**
**Changes in ST-segment** ^‡^	44 (64.7)	56 (68.3)	0.772
**Left ventricular ejection fraction (%)** ^‡^			
>50	31 (45.6)	49 (59.8)	0.158
30–50	31 (45.6)	30 (36.6)	
<30	6 (8.8)	3 (3.7)	

^‡^: *n* (%), ^†^: mean ± standard deviation, ^§^: median [min–max]; PCI: percutaneous coronary intervention, CABG: coronary artery bypass grafting.

**Table 3 medicina-62-00036-t003:** Risk assessment of the groups based on the Killip classification and Grace and SYNTAX risk scores.

	Group PCI (*n* = 68)	Group CABG (*n* = 82)	*p*
**Killip Class 3–4** ^‡^	4 (5.9)	10 (12.2)	0.298
**GRACE score** ^†^	122.7 ± 20.1	112.3 ± 23.7	**0.004**
**GRACE score categories** ^‡^			
<109	18 (26.5)	38 (46.3)	**0.034**
109–140	38 (55.9)	36 (43.9)	
>140	12 (17.6)	8 (9.8)	
**SYNTAX score** ^§^	16.0 [5.0–45.0]	24.0 [10.0–45.0]	**<0.001**
**SYNTAX score categories** ^‡^			
0–22	52 (76.5)	35 (43.2)	**<0.001**
23–32	10 (14.7)	36 (44.4)	
≥33	6 (8.8)	10 (12.3)	

^‡^: *n* (%), ^†^: mean ± standard deviation, ^§^: median [min–max]; PCI: percutaneous coronary intervention, CABG: coronary artery bypass grafting.

**Table 4 medicina-62-00036-t004:** In-hospital outcomes and the prespecified in-hospital composite endpoint.

	Group PCI (*n* = 68)	Group CABG (*n* = 82)	*p*
**In-hospital outcomes**			
In-hospital composite endpoint ^‡^	27 (39.7)	32 (39.0)	0.999
Reinfarction ^‡^	2 (2.9)	1 (1.2)	0.590
Major bleeding ^‡^	2 (2.9)	17 (20.7)	**0.003**
Acute heart failure ^‡^	4 (5.9)	8 (9.8)	0.570
Acute renal failure ^‡^	24 (35.3)	17 (20.7)	0.071
Length of stay (day) ^§^	6.0 [2.0–24.0]	21.0 [7.0–99.0]	**<0.001**
Mortality ^‡^	2 (2.9)	10 (12.2)	0.075

^‡^: *n* (%), ^§^: median [min–max]; PCI: percutaneous coronary intervention, CABG: coronary artery bypass grafting.

**Table 5 medicina-62-00036-t005:** Long-term outcomes of the study groups.

	Group PCI (*n* = 68)	Group CABG (*n* = 82)	*p*
Duration for follow-up (day) ^§^	517.0 [2.0–840.0]	490.5 [11.0–1341.0]	0.120
Major adverse cardiac event ^‡^	21 (30.9)	18 (22.0)	0.292
Nonfatal myocardial infarction ^‡^	14 (20.6)	9 (11.0)	0.162
Revascularization ^‡^	13 (19.1)	5 (6.1)	**0.028**
Need for hemodialysis ^‡^	9 (13.2)	9 (11.0)	0.864
Mortality ^‡^	3 (4.4)	2 (2.4)	0.659

^‡^: *n* (%), ^§^: median [min–max]; PCI: percutaneous coronary intervention, CABG: coronary artery bypass grafting.

## Data Availability

The original contributions presented in this study are included in the article. Further inquiries can be directed to the corresponding author.

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
