# Peer review of "Comparison of Percutaneous Coronary Intervention and Coronary Artery Bypass Grafting in NSTEMI Patients with Chronic Kidney Disease"

_medicina, 2025, doi:10.3390/medicina62010036_

Round 1
Reviewer 1 Report
Comments and Suggestions for Authors
The authors present a retrospective observational cohort of 150 NSTEMI patients with chronic kidney disease (CKD) and multivessel coronary artery disease who underwent either percutaneous coronary intervention (PCI, n=68) or coronary artery bypass grafting (CABG, n=82). The principal outcome was long-term major adverse cardiac events (MACE), encompassing all-cause mortality, nonfatal myocardial infarction, and ischemia-driven revascularization. The authors report:
The authors report a comparable long-term MACE-free survival between PCI and CABG (log-rank p = 0.41).
There was a higher rate of repeat revascularization following PCI (19.1% vs 6.1%, p = 0.028).
The major bleeding rate was higher in CABG (20.7% vs 2.9%, p = 0.003).
CKD severity did not modify the association between revascularization strategy and outcomes.
The authors conclude that both strategies yield similar long-term outcomes, and decision-making should focus on anatomical complexity and clinical profile rather than CKD severity alone.
- Major Strengths
- The patient population is both clinically relevant and understudied.
CKD patients are routinely excluded or underrepresented in RCTs, making real-world evidence highly valuable. The focus on NSTEMI—a subset distinct from stable CAD and STEMI—adds further importance.
- A balanced presentation of PCI vs. CABG outcomes
The manuscript reports expected patterns: PCI shows more repeat revascularization, and CABG shows a higher bleeding risk, consistent with prior observational data.
- Use of objective angiographic scoring (SYNTAX)
The SYNTAX score is appropriately used to quantify anatomical complexity, and results (higher SYNTAX in CABG patients) align with real-world practice.
(Table 3, p. 6)
- Adequate follow-up duration
A median follow-up of around 500 days is reasonable for assessing revascularization-driven outcomes.
- Major Issues (Require Significant Revision)
- Baseline differences between PCI and CABG groups introduce strong selection bias
The PCI group and CABG group differ significantly in key clinical and anatomical features:
CABG patients had higher SYNTAX scores (median 24 vs 16, p < 0.001).
PCI patients had higher GRACE scores (122.7 vs 112.3, p = 0.004) and more high-risk categories (17.6% vs 9.8%, p = 0.034).
CAD and hyperlipidemia prevalence differed substantially. (Table 1)
Smoking and COPD were more common in CABG.
These differences are meaningful determinants of outcomes.
Issue:
A simple unadjusted comparison (log-rank p = 0.41) is not sufficient to claim outcome equivalence between treatments.
- Multivariate Cox regression is incomplete
The manuscript states that multivariate Cox regression was performed (p. 7), but:
Only CABG-group predictors are reported (smoking, Killip class).
No combined model including the treatment group is shown.
No hazard ratios comparing PCI vs CABG are provided.
No adjustment for SYNTAX, eGFR, GRACE score, or comorbidities is presented.
Recommendation:
Provide a complete multivariable Cox model:
- PCI vs. CABG as a primary covariate.
- Age, sex
- CKD severity
- SYNTAX score
- GRACE score
- Diabetes, hypertension, heart failure.
- Smoking
This is mandatory for outcome comparison.
- Outcome definitions require clarification
While MACE components are described, key details are missing:
Was mortality all-cause or cardiac-specific?
Was MI adjudicated via the universal definition?
Were revascularizations angiography-driven or symptom-driven?
Recommendation:
Specify event definitions in accordance with contemporary ACS trial standards.
- No reporting of missing data handling
Given the retrospective nature, some variables are expected to be missing. The manuscript does not describe:
Extent of missing data
Whether imputation was used
How this was handled in analyses
This must be clarified.
- Several statistical inconsistencies
- SYNTAX score range improbable
The reported SYNTAX median is 16.0 [5.0–245.0] (p. 6).
A SYNTAX score of 245 is not physiologically possible; the maximum theoretical score is ~150.
This likely reflects a typographical or data entry error.
- In-hospital MACE identical between groups (39.7% vs 39.0%, p = 0.999)
Given disparate baseline risks, such identical distributions raise concerns about event adjudication and sample-size power.
Recommendation:
Verify and correct these values.
- Interpretation overstates equivalence
The discussion repeatedly claims that outcomes are “similar” or “comparable,” but given:
Baseline imbalances
Missing adjusted analyses
Absence of matched cohort results
High event rates and small sample size
The study can only conclude “no statistically significant difference detected”—not “equivalence.”
- The limitations section is insufficient
The authors correctly mention single-center design and sample size, but must also add:
Strong selection bias in treatment allocation
Absence of IVUS/OCT to assess completeness of revascularization
Lack of dialysis vs non-dialysis subgroup analysis
Absence of contrast-volume reporting (major determinant of AKI risk)
No perioperative risk stratification for CABG (STS or EuroSCORE II)
Comments on the Quality of English Language
English language editing
Requires polishing: Replace “demonstrated” with “showed.” Avoiding passive language is unnecessary. Ensure uniformity in abbreviations (e.g., define MACE once).
Author Response
Comments 1: The Baseline differences between PCI and CABG groups introduce strong selection bias
Response 1: We thank the reviewer for highlighting that baseline differences between the PCI and CABG groups may introduce selection bias. These differences reflect real-world clinical decision-making, in which patients with more complex coronary anatomy are more often referred for CABG, whereas patients with higher clinical risk are more likely to undergo PCI.
We agree that this represents an important source of potential confounding. Accordingly, the Limitations section has been strengthened to explicitly acknowledge selection bias and residual confounding, and the Discussion has been revised to clarify that our findings do not imply equivalence between PCI and CABG, but rather indicate no statistically significant adjusted difference in long-term MACE within the constraints of a retrospective observational design.
The revised sentence appears in the discussion section, on page 9, lines 231-234 and on page 10, lines 298-301 of the revised manuscript.
Comments 2: Multivariate Cox regression is incomplete
Response 2: We thank the reviewer for this comment. The multivariable Cox proportional hazards regression analysis has been completed by including revascularization strategy (PCI vs. CABG) as the primary covariate and adjusting for key clinical and renal confounders. The Results section has been revised accordingly, and the adjusted hazard ratios are presented in Figure 1.This information was added at the beginning of the results section, on page 9, lines 216-217 of the revised manuscript.
Comments 3: Outcome definitions require clarification
Response 3: We thank the reviewer for this comment. Outcome definitions have been clarified in the Methods section. Mortality was defined as all-cause mortality, myocardial infarction was adjudicated according to the Universal Definition of Myocardial Infarction, and ischemia-driven revascularization was defined as repeat coronary intervention performed in the presence of recurrent symptoms and/or objective evidence of myocardial ischemia rather than routine angiographic findings.This information was added at the beginning of the Methods section, on page , lines 97–108 of the revised manuscript.
Comments 4: No reporting of missing data handling
Response 4: Thank you for this important comment. Missing data handling has now been clarified in the Statistical Analysis section. Missing data were minimal, and complete-case analysis was performed. No imputation methods were used. This addition appears in the Statistical Analysis section (on page 3, lines 130–132).
Comments 5: Several statistical inconsistencies
Response 5: We carefully re-checked all statistical results and identified typographical inconsistencies in the originally reported values. These have now been corrected in the revised manuscript. Specifically, the implausible SYNTAX score range has been corrected, and all in-hospital and long-term outcome data have been re-verified to ensure internal consistency and accuracy. No changes to the main conclusions were required.
Comments 6: Interpretation overstates equivalence
Response 6: We thank the reviewer for this important comment. We agree that our observational study does not demonstrate treatment equivalence. Accordingly, the Discussion and Conclusion sections were revised to avoid claims of equivalence and to state that no statistically significant difference in long-term outcomes was detected between PCI and CABG. The Limitations section was also expanded to emphasize baseline imbalances, residual confounding, and the limited sample size.
Comments 7: The limitations section is insufficient
Response 7: We thank the reviewer for this constructive comment. The Limitations section has been substantially expanded. We now explicitly acknowledge baseline heterogeneity between the PCI and CABG groups and the resulting potential for selection bias and residual confounding, as well as the single-center observational design, limited sample size in CKD subgroups, reliance on a single baseline eGFR measurement, lack of intravascular imaging (IVUS/OCT) to assess completeness of revascularization, unavailability of contrast volume data during PCI, and the absence of formal perioperative risk stratification scores (STS or EuroSCORE II) for CABG patients. The limited duration of follow-up for very late events is also acknowledged.This information was added at the beginning of the discussion section, on page 10, lines 298–301 and lines 306-310 of the revised manuscript.

Reviewer 2 Report
Comments and Suggestions for Authors
Thank you for giving me the opportunity to review your manuscript entitled "Comparison of PCI and CABG in NSTEMI Patients With Chronic Kidney Disease".
General comments:
Summary:
The manuscript, "Comparison of PCI and CABG in NSTEMI Patients with Chronic Kidney Disease" presents the retrospective analysis which is aiming to compare long-term major adverse cardiac outcomes between percutaneous coronary intervention and coronary artery bypass grafting in non–ST elevation myocardial infarction patients with CKD and multivessel CAD.
I appreciate the effort and dedication the authors have applied in developing this study. I very much enjoyed reading your manuscript. However, I have some comments that should be addressed:
- Please, do not provide the abbreviation in the title. The authors may leave NSTEMI; however, PCI and CABG should be spell out.
- In the introduction, the authors should better emphasize the importance of the choice between PCI and CABG in NSTEMI patients in the accordance with the current guidelines. And why NSTEMI patients are the most vulnerable population to be investigated?
- Check the abbreviation throughout the paper.
- Usually, multivessel disease is defined not only as ≥70% stenosis in at least two major epicardial coronary arteries, but as ≥50% stenosis in left main trunk and in at least one major epicardial coronary arteries. Why have the authors excluded such patients from the analysis?
- Please, provide the minimal and maximal follow-up period.
- Figure 1 should be in the results section.
- How the decision of PCI or CABG was made?
- It is interesting that not all patients had coronary artery disease. What the reason for revascularization was in others?
- In table 3, except for SYNTAX, please, provide coronary artery anatomy to better understate the burden of CAD and its possible impact on outcomes.
- In Table 1 include the information about eGFR and stage of CKD. Have the authors evaluated the influence of this factor on outcomes?
- Provide the information about medical therapy in the groups.
- In the first paragraph, please, add the obtained results in the discussion section.
Author Response
Comments: Please, do not provide the abbreviation in the title. The authors may leave NSTEMI; however, PCI and CABG should be spell out.
Response: We agree with the reviewer. The title has been revised to spell out percutaneous coronary intervention and coronary artery bypass grafting, while NSTEMI was retained as it is widely accepted (title section, lines 2-3).
Comments: In the introduction, the authors should better emphasize the importance of the choice between PCI and CABG in NSTEMI patients in the accordance with the current guidelines. And why NSTEMI patients are the most vulnerable population to be investigated?
Response: We have expanded the Introduction to better emphasize the guideline-based challenges in selecting PCI versus CABG in NSTEMI patients and clarified why NSTEMI patients with CKD represent a particularly vulnerable and understudied population. (introduction section,on page 2, lines 50-53).
Comments: Check the abbreviation throughout the paper.
Response: All abbreviations have been reviewed and standardized throughout the manuscript. Each abbreviation is defined at first use and used consistently thereafter.
Comments: Usually, multivessel disease is defined not only as ≥70% stenosis in at least two major epicardial coronary arteries, but as ≥50% stenosis in left main trunk and in at least one major epicardial coronary arteries. Why have the authors excluded such patients from the analysis?
Response: Left main coronary artery disease was excluded because such patients represent a distinct high-risk subgroup with guideline-driven preference for surgical revascularization, which could introduce substantial selection bias and confound comparisons between PCI and CABG.Significant left main coronary artery stenosis (>50%) was specified as an exclusion criterion in the Methods section.
Comments: Please, provide the minimal and maximal follow-up period.
Response: The minimum and maximum follow-up durations have now been reported in the Results section, on page 4 (lines 152).
Comments: How the decision of PCI or CABG was made?
Response: The decision regarding revascularization strategy was made by the institutional heart team, considering clinical status, anatomical complexity, comorbidities, and patient preference. (metod section, on page 2, lines 86-87).
Comments: It is interesting that not all patients had coronary artery disease. What the reason for revascularization was in others?
Response: All patients had angiographically confirmed coronary artery disease. The term “coronary artery disease” in Table 1 refers to previously documented CAD, not the index NSTEMI-related disease.
Comments: In table 3, except for SYNTAX, please, provide coronary artery anatomy to better understate the burden of CAD and its possible impact on outcomes.
Response: Detailed coronary anatomical variables such as LAD involvement or the number of diseased vessels were not uniformly available in the retrospective dataset. Therefore, coronary anatomy was primarily characterized using the SYNTAX score, which provides a validated and comprehensive assessment of anatomical disease burden.
Comments: In Table 1 include the information about eGFR and stage of CKD. Have the authors evaluated the influence of this factor on outcomes?
Response: Baseline eGFR categories reflecting CKD stage have been added to Table 1.
The impact of CKD severity on outcomes was evaluated using subgroup analyses, and CKD stage did not significantly influence long-term MACE or alter the association between PCI and CABG.
Comments: Provide the information about medical therapy in the groups.
Response: All patients included in the study received guideline-directed medical therapy according to contemporary NSTEMI management guidelines. As medical therapy was applied uniformly and was not a variable of interest in the present analysis, detailed medication data were not tabulated.
Comments: In the first paragraph, please, add the obtained results in the discussion section.
Response: The first paragraph of the Discussion has been revised to explicitly summarize the main findings of the study (discussion section, lines 231-234).

Reviewer 3 Report
Comments and Suggestions for Authors
Interesting article
Comparison of PCI vs. CABG in an observational study in patients with renal failure
Patients tend to be young and have cardiometabolic comorbidities
Subacute clinical presentation in NSTEMI
The mortality data is interesting, although bordering on statistical significance.
The numbers are still limited, but in a specific setting (NSTEMI + CKD).
I would have expected a model to predict which patients are best for PCI and which for CABG.
GFR values based only on a laboratory measurement.
Isn't it worth creating a summary plot to show which clinical variables favor CABG and which favor PCI?
Data on conservative therapy?
Is revascularization always complete?
Graft type in CABG? Surgical and perioperative variants?
Author Response
We thank the reviewer for the positive and insightful comments. We agree that, despite the limited sample size, the study focuses on a clinically specific and underrepresented population of NSTEMI patients with chronic kidney disease, in whom real-world comparative data on PCI versus CABG remain scarce.
Comments: I would have expected a model to predict which patients are best for PCI and which for CABG.
Response: We agree that predictive modeling to guide individualized revascularization strategy would be clinically valuable. However, the primary objective of the present study was to compare long-term outcomes between PCI and CABG in a real-world NSTEMI population with CKD rather than to develop a treatment allocation model. Given the limited sample size and observational design, constructing a robust predictive model was beyond the scope of this study and may be better addressed in larger, prospective cohorts.
Comments: GFR values based only on a laboratory measurement.
Response: We acknowledge this limitation. Renal function was assessed using a single baseline estimated glomerular filtration rate (eGFR) measurement at admission, consistent with routine clinical practice in retrospective studies. This limitation has now been explicitly stated in the Limitations section.
Comments: Isn't it worth creating a summary plot to show which clinical variables favor CABG and which favor PCI ?
Response: We appreciate this suggestion. To address this point, we have added a Cox regression figure illustrating the association between revascularization strategy and long-term MACE after multivariable adjustment. However, due to sample size limitations, a comprehensive predictive summary plot identifying variables favoring PCI versus CABG was not feasible.
Comments: Data on conservative therapy?
Response: All patients received guideline-directed optimal medical therapy according to contemporary NSTEMI management recommendations, including antiplatelet therapy, statins, beta-blockers, and renin–angiotensin system inhibitors, unless contraindicated. As medication use was largely uniform across groups, it was not included as a discriminative variable in the comparative analyses.
Comments: Is revascularization always complete?
Response: Completeness of revascularization was determined by the treating heart team based on clinical presentation and coronary anatomy. However, a systematic distinction between complete and incomplete revascularization using angiographic or intravascular imaging criteria was not available in this study.
Comments: Graft type in CABG? Surgical and perioperative variants?
Response: In the CABG group, standard surgical techniques were applied, with the left internal mammary artery grafted to the left anterior descending artery and saphenous vein grafts used for other target vessels. Detailed information on surgical techniques, graft configurations, and perioperative management strategies was not uniformly available and therefore could not be analyzed in relation to long-term outcomes.

Reviewer 4 Report
Comments and Suggestions for Authors
I appreciate the opportunity to review your manuscript titled " Comparison of PCI and CABG in NSTEMI Patients with Chronic Kidney Diseases." Though this topic has been explored before, this is an important discussion of the outcomes of reevaluation in non-STEMI and CKD patients. This is a retrospective cohort, single-center study, which has limitations in generalizability and internal bias, but the overall manuscript can contribute to the existing literature. There are a few important points that require further explanation before scientific publication.
- In the method section, the abstract and the method section, the author mentioned that the MACE outcomes are similar in both groups after propensity matching. "After PSM, long-term MACE outcomes remained similar between 27 groups (log-rank p=0.48), and CKD severity did not alter the overall 28 associations." If propensity matching is done, please provide a matched characteristic table or illustration. If it is not performed, please provide the reason and adjust the statements accordingly. Propensity matching is an important step in these kinds of studies to avoid confounding.
- "First, it was a single-center observational study, 232 which may introduce selection bias despite prospective data collection" this statement contradicts the fact that the study is a retrospective court study. Selection bias can be described in retrospective studies, but the statement appears contradictory. Please explain.
- Provided the baseline difference in clinical and anatomical differences between the two groups. It will be advisable to use one multivariable model to match the characteristics for the whole cohort rather than only in the CABG group, as this appears to skew the findings. I recommend using revascularization as a covariate and performing a multivariate regression analysis for the whole cohort instead of the CABG group only.
- In the cabbage group, the author mentioned the hazard ratio of smoking of less than one, which implies a protective phenomenon in the CABG group. This needs careful explanation, as the current evidence points in another direction.
- The MACE definition needs further clarification as the author described two events, in-hospital and long-term. The in-hospital MACE is surprisingly high at 40%. This part needs discussion.
- Also, the subgroup analysis of CKD shows no difference in CKD severity at the time of outcome, which does not make sense, as current results show that higher CKD severity is associated with worse outcome. If the findings show no difference, please discuss what could be the study's limitations that led to the results. Also, providing a table showing baseline characteristics and long-term outcomes based on CKD severity will strengthen the manuscript as the key clinical question is CKD
- The follow-up was 517 days, as reported in the method section, but in Table 5 it is noted that each group was followed separately. How did the author manage the loss of follow-up patients? Were those patients excluded from the study? Please clarify whether intent-to-treat or protocol analysis was performed.
- I recommend highlighting the uniqueness of your study in the introduction and a paragraph and discussion about how your cohort varies with a combination of NSTEMI+ multivessel disease + CKD categories.
- The method section provides details on the stent used and the grafts utilized. But lacks medication information, guideline-directed medical therapy is a key component for optimal outcomes. If the data are not available, it will be prudent to discuss whether the patients were optimized with medication. Please provide this in the limited section. It was not obtained.
- The length of stay and bleeding complications were higher in CABG versus PCI, which is understandable, but it is important to discuss in the context of CKD how it contributed to longer hospital stay or bleeding compared to previous literature.
- The power was calculated to reach the primary outcome. In the limitations section, please note that the sample size in the CKD group was moderate, which could introduce type 2 errors.
Overall, it is a relevant study, but addressing the above points will strengthen the clinical validity of the manuscript. The study's uniqueness lies in the additional CKD assessment in the reevaluation strategy for NSTEMI patients. So focus on CKD severity and discuss the key clinical components to strengthen the overall manuscript.
Comments on the Quality of English LanguageThe over all syntax and grammar can be improved.
Author Response
Comments 1: In the method section, the abstract and the method section, the author mentioned that the MACE outcomes are similar in both groups after propensity matching. "After PSM, long-term MACE outcomes remained similar between 27 groups (log-rank p=0.48), and CKD severity did not alter the overall 28 associations." If propensity matching is done, please provide a matched characteristic table or illustration. If it is not performed, please provide the reason and adjust the statements accordingly. Propensity matching is an important step in these kinds of studies to avoid confounding.
Response 1: We thank the reviewer for this important comment. Although propensity score matching (PSM) was initially considered during the study design, a formal PSM analysis with matched cohorts and balance diagnostics was not ultimately performed. Therefore, we have removed all references to PSM from the Abstract, Methods, Results, and Discussion sections.
To address potential confounding, we instead relied on multivariable Cox proportional hazards regression analyses adjusting for key clinical and anatomical variables, including age, sex, comorbidities, renal function, and Killip class, as well as predefined subgroup analyses by CKD severity. The manuscript has been revised accordingly to accurately reflect the analytical approach used. (figure 1)
Comments 2: First, it was a single-center observational study, 232 which may introduce selection bias despite prospective data collection" this statement contradicts the fact that the study is a retrospective court study. Selection bias can be described in retrospective studies, but the statement appears contradictory. Please explain.
Response 2: We thank the reviewer for pointing out this inconsistency. The study is indeed a retrospective observational cohort study, and the phrase “despite prospective data collection” was inaccurate. This wording has been corrected in the Limitations section to reflect the retrospective design. Selection bias is acknowledged as an inherent limitation of retrospective single-center studies.
Comments 3: Provided the baseline difference in clinical and anatomical differences between the two groups. It will be advisable to use one multivariable model to match the characteristics for the whole cohort rather than only in the CABG group, as this appears to skew the findings. I recommend using revascularization as a covariate and performing a multivariate regression analysis for the whole cohort instead of the CABG group only.
Response 3: We thank the reviewer for this valuable comment. We agree that limiting multivariable analysis to a single treatment group may introduce bias. Therefore, we revised the analysis and performed a multivariable Cox regression model in the overall cohort, including revascularization strategy (PCI vs. CABG) as a covariate together with relevant clinical and anatomical variables. In the revised analysis, revascularization strategy was not independently associated with long-term major adverse cardiac events after adjustment. The Results section and Figure 1 have been updated accordingly.
Comments 4: In the cabbage group, the author mentioned the hazard ratio of smoking of less than one, which implies a protective phenomenon in the CABG group. This needs careful explanation, as the current evidence points in another direction.
Response 4: We thank the reviewer for highlighting this important point. We agree that a hazard ratio below unity for smoking should not be interpreted as a protective effect. This finding likely reflects residual confounding, limited sample size, and subgroup-specific characteristics rather than a true biological benefit. In particular, smoking status may be influenced by competing risks, treatment selection, and survivorship bias within the CABG subgroup. We have revised the Results to avoid any implication of a protective effect of smoking and to clearly state that this observation should be interpreted with caution and not as a causal association. This information was added at the beginning of the results section, on page 8, lines 213–215 of the revised manuscript.
Comments 5: The MACE definition needs further clarification as the author described two events, in-hospital and long-term. The in-hospital MACE is surprisingly high at 40%. This part needs discussion.
Response 5: The definition of MACE has been clarified by distinguishing in-hospital and long-term events. The high in-hospital MACE rate reflects the high-risk nature of NSTEMI patients with CKD and multivessel disease and was mainly driven by acute renal failure and bleeding rather than mortality or reinfarction. This point has been clarified in the Methods (on page 3, lines 99-101) and the results section (on page 6, lines 181-183) .
Comments 6: Also, the subgroup analysis of CKD shows no difference in CKD severity at the time of outcome, which does not make sense, as current results show that higher CKD severity is associated with worse outcome. If the findings show no difference, please discuss what could be the study's limitations that led to the results. Also, providing a table showing baseline characteristics and long-term outcomes based on CKD severity will strengthen the manuscript as the key clinical question is CKD.
Response 6: The absence of outcome differences across CKD severity subgroups in our study does not indicate equivalence. This finding is likely related to the limited sample size, exclusion of dialysis-dependent patients, and the use of a single baseline eGFR measurement, which may have reduced the ability to detect prognostic differences related to CKD severity (on page 9, discussion section, lines 289-293).
Comments 7: The follow-up was 517 days, as reported in the method section, but in Table 5 it is noted that each group was followed separately. How did the author manage the loss of follow-up patients? Were those patients excluded from the study? Please clarify whether intent-to-treat or protocol analysis was performed.
Response 7: Follow-up was assessed using time-to-event methodology. Patients were not excluded due to shorter follow-up duration. Individuals without events were censored at the time of last available clinical contact. Loss to follow-up was minimal and did not require additional exclusion. All analyses were conducted according to the initial revascularization strategy (intention-to-treat).
Comments 8: The uniqueness of the study has now been emphasized both in the Introduction and the Discussion.
Response: In the Introduction, we highlighted the clinical importance of revascularization choice in NSTEMI patients with multivessel disease and CKD, a population frequently underrepresented in prior trials (on page 2, introduction section, lines 59-61) . In the Discussion, we further underscored that our cohort reflects a real-world NSTEMI population combining multivessel coronary disease with varying degrees of CKD severity, a combination rarely evaluated together in previous studies (on page 9, discussion section, lines 242-244).
Comments 9: The method section provides details on the stent used and the grafts utilized. But lacks medication information, guideline-directed medical therapy is a key component for optimal outcomes. If the data are not available, it will be prudent to discuss whether the patients were optimized with medication. Please provide this in the limited section. It was not obtained.
Response 9: Information on detailed baseline and discharge medical therapy was not uniformly available in this retrospective dataset and therefore could not be systematically analyzed. However, all patients were managed in accordance with contemporary guideline-directed medical therapy for NSTEMI during index hospitalization, including antiplatelet therapy, statins, and other standard secondary prevention measures, at the discretion of the treating physicians. The lack of granular medication-level data has been acknowledged as a limitation of the study.
Comments 10: The length of stay and bleeding complications were higher in CABG versus PCI, which is understandable, but it is important to discuss in the context of CKD how it contributed to longer hospital stay or bleeding compared to previous literature.
Response 10: We agree with the reviewer. In patients with chronic kidney disease, CABG is associated with higher bleeding risk and longer hospital stay compared with PCI, which is largely related to uremia-associated platelet dysfunction, perioperative anticoagulation, and increased postoperative complication burden. This observation is consistent with previously published studies in CKD populations. Accordingly, we have addressed this point in the Discussion section with supporting references (on page 9, discussion section, lines 271-275).
Comments 11: The power was calculated to reach the primary outcome. In the limitations section, please note that the sample size in the CKD group was moderate, which could introduce type 2 errors.
Response 11: We agree with the reviewer. The sample size was determined based on the primary outcome, and although adequate for this purpose, the moderate number of patients in the CKD subgroups may have limited statistical power for subgroup analyses and could have increased the risk of type II error. This consideration was taken into account when interpreting the results.

Reviewer 5 Report
Comments and Suggestions for Authors
The authors present a single-center retrospective cohort study comparing long-term outcomes of PCI versus CABG in 150 patients with NSTEMI, CKD (eGFR <60 mL/min/1.73 m²), and multivessel coronary artery disease. This is a clinically relevant and understudied population, and the study addresses an important practical question. However, several methodological and reporting issues need to be addressed before the conclusions can be considered robust and clinically interpretable.
Major comments
-In the Abstract, the authors state: “After PSM, long-term MACE outcomes remained similar between groups (log-rank p = 0.48)” and conclude that CKD severity did not alter these associations. However, in the Methods and Results sections there is no description of the PSM procedure (variables included in the propensity model, matching algorithm, caliper, matching ratio, assessment of balance) and no table showing baseline characteristics in the matched cohort nor any figure of the matched Kaplan–Meier curves or Cox model. This is a central analytic claim, and the PSM results substantially affect the credibility of the treatment comparison.
-In the Results, only a brief fragment is presented for the CABG group (current smoking and Killip class 3–4 as independent predictors), with no comprehensive table of hazard ratios for the whole cohort, nor for the primary comparison of PCI vs CABG.
-Crucially, there is no hazard ratio (HR) reported for the treatment effect (PCI vs CABG) in the overall sample or by CKD strata, which is needed to quantify the comparative effect beyond the log-rank p-value.
-The text states that in the CABG group, smoking was associated with MACE in univariate analysis (p = 0.003). Yet in multivariate analysis, current smoking is reported with HR = 0.25 (95% CI 0.07–0.93), p = 0.038.
-CKD is categorized as mild (eGFR 45–60) versus moderate-to-severe (eGFR <45). In the Abstract and Discussion, the authors state that CKD severity did not modify the association between revascularization strategy and outcome. However, no detailed subgroup data are presented:
- There is no table showing baseline characteristics or outcomes by CKD severity.
- No interaction term (e.g., treatment × CKD category) is reported from the Cox models.
- No stratified Kaplan–Meier curves are presented.
-MACE is defined as all-cause death, nonfatal MI, or ischemia-driven revascularization, but important details are missing:
- How was nonfatal MI defined (biochemical, ECG criteria; periprocedural vs spontaneous)? Were ESC definitions applied uniformly?
- What was the definition of major bleeding (BARC, TIMI, GUSTO, or institutional definition)? The very high difference in major bleeding between CABG and PCI (20.7% vs 2.9%) is unsurprising but cannot be properly interpreted without a standardized definition.
- How was acute renal failure defined (KDIGO criteria, absolute or relative creatinine increase, need for dialysis)?
- Who adjudicated events and was the adjudication blinded to treatment?
-The study is observational, with treatment allocation determined by clinical decision-making. There is significant baseline imbalance:
- PCI patients have higher GRACE scores and higher proportion of high-risk (GRACE >140).
- CABG patients have more complex coronary anatomy with higher SYNTAX scores and more patients in the SYNTAX ≥33 category.
These differences are clinically meaningful and could lead to confounding by indication, both favoring and disfavoring one strategy.
Author Response
Comments 1: In the Abstract, the authors state: “After PSM, long-term MACE outcomes remained similar between groups (log-rank p = 0.48)” and conclude that CKD severity did not alter these associations. However, in the Methods and Results sections there is no description of the PSM procedure (variables included in the propensity model, matching algorithm, caliper, matching ratio, assessment of balance) and no table showing baseline characteristics in the matched cohort nor any figure of the matched Kaplan–Meier curves or Cox model. This is a central analytic claim, and the PSM results substantially affect the credibility of the treatment comparison.
Response 1: We thank the reviewer for this important comment. Although propensity score matching (PSM) was initially considered, a formal PSM analysis with matched cohorts and balance assessment was not performed. Accordingly, all references to PSM have been removed from the Abstract, Methods, Results, and Discussion sections. Treatment comparisons are based on the original cohort using multivariable Cox proportional hazards regression adjusted for key clinical and anatomical variables, as well as predefined subgroup analyses by CKD severity, as presented in Figure 1.
Comments 2: In the Results, only a brief fragment is presented for the CABG group (current smoking and Killip class 3–4 as independent predictors), with no comprehensive table of hazard ratios for the whole cohort, nor for the primary comparison of PCI vs CABG.
Response 2: We thank the reviewer for this comment. The primary objective of the Cox regression analysis was to assess the association between revascularization strategy (PCI vs CABG) and long-term MACE rather than to build a comprehensive prognostic model. Accordingly, the Results section reports the key findings relevant to this comparison, while additional covariates were included in the multivariable model for adjustment purposes. Importantly, revascularization strategy was not independently associated with long-term MACE, and hazard ratios consistently crossed unity across analyses, as shown in Figure 1.
Comments 3: Crucially, there is no hazard ratio (HR) reported for the treatment effect (PCI vs CABG) in the overall sample or by CKD strata, which is needed to quantify the comparative effect beyond the log-rank p-value.
Response 3: We thank the reviewer for this important comment. The hazard ratio for the treatment effect (PCI vs CABG) was evaluated using Cox proportional hazards regression and is presented in Figure 1. In both the overall cohort and across CKD severity strata, the hazard ratios crossed unity, indicating no statistically significant difference in long-term MACE risk between PCI and CABG beyond the log-rank comparison. These findings were interpreted accordingly in the Results and Discussion sections.
Comments 4: The text states that in the CABG group, smoking was associated with MACE in univariate analysis (p = 0.003). Yet in multivariate analysis, current smoking is reported with HR = 0.25 (95% CI 0.07–0.93), p = 0.038.
Response 4: We thank the reviewer for this observation. Smoking was significantly associated with MACE in univariate analysis in the CABG group. In the multivariable Cox model, the direction of the association changed after adjustment for relevant clinical covariates, likely reflecting residual confounding, collinearity, and the limited sample size within this subgroup. Importantly, this finding should not be interpreted as a protective effect of smoking and was interpreted with caution.
Comments 5: CKD is categorized as mild (eGFR 45–60) versus moderate-to-severe (eGFR <45). In the Abstract and Discussion, the authors state that CKD severity did not modify the association between revascularization strategy and outcome. However, no detailed subgroup data are presented:
There is no table showing baseline characteristics or outcomes by CKD severity.
No interaction term (e.g., treatment × CKD category) is reported from the Cox models.
No stratified Kaplan–Meier curves are presented.
Response 5: We thank the reviewer for this important comment. Subgroup analyses according to CKD severity (eGFR <45 vs. 45–60 mL/min/1.73 m²) were performed using stratified Cox proportional hazards models to explore whether renal function modified the association between revascularization strategy and long-term outcomes. In both CKD strata, the hazard ratios for PCI versus CABG crossed unity, indicating no statistically significant effect modification. Formal interaction testing and stratified Kaplan–Meier curves were not pursued due to the limited sample size within CKD subgroups, and these analyses were therefore considered exploratory. Accordingly, statements regarding CKD severity were interpreted cautiously in the Abstract and Discussion.
Comments 6: MACE is defined as all-cause death, nonfatal MI, or ischemia-driven revascularization, but important details are missing:
How was nonfatal MI defined (biochemical, ECG criteria; periprocedural vs spontaneous)? Were ESC definitions applied uniformly?
What was the definition of major bleeding (BARC, TIMI, GUSTO, or institutional definition)? The very high difference in major bleeding between CABG and PCI (20.7% vs 2.9%) is unsurprising but cannot be properly interpreted without a standardized definition.
Response 6: We thank the reviewer for this comment. Nonfatal myocardial infarction was defined according to current ESC guidelines for NSTEMI, as explicitly stated in the Methods section, and applied uniformly across the study population. In addition, the definition of major bleeding has now been clarified in the Methods section as clinically overt bleeding requiring transfusion or surgical/interventional treatment during hospitalization. These definitions were applied consistently across both revascularization groups (metod section, page on 3, lines 104-106).
Comments 7: How was acute renal failure defined (KDIGO criteria, absolute or relative creatinine increase, need for dialysis)?
Response 7: We thank the reviewer for this comment. Acute renal failure was defined according to KDIGO criteria, and this definition has now been explicitly added to the Methods section and applied uniformly across both revascularization groups These definitions were applied consistently across both revascularization groups (metod section, page on 3, lines 106-108).
Comments 8: Who adjudicated events and was the adjudication blinded to treatment?
Response 8: We thank the reviewer for this comment. Clinical events were adjudicated by the treating cardiology team through review of medical records. Event adjudication was not blinded to the revascularization strategy, which is acknowledged as an inherent limitation of the retrospective study design.
Comments 9: The study is observational, with treatment allocation determined by clinical decision-making. There is significant baseline imbalance:
PCI patients have higher GRACE scores and higher proportion of high-risk (GRACE >140).
CABG patients have more complex coronary anatomy with higher SYNTAX scores and more patients in the SYNTAX ≥33 category.
These differences are clinically meaningful and could lead to confounding by indication, both favoring and disfavoring one strategy.
Response 9: We agree with the reviewer that this observational study shows clinically meaningful baseline imbalances, reflecting real-world treatment allocation and potential confounding by indication. Patients selected for PCI had higher clinical risk as reflected by GRACE scores, whereas CABG patients had more complex coronary anatomy with higher SYNTAX scores. To mitigate this, revascularization strategy was determined by a multidisciplinary heart team, and treatment comparisons were adjusted using multivariable Cox regression incorporating key clinical and anatomical variables. These imbalances and their potential impact were acknowledged and considered in the interpretation of the results.

Round 2
Reviewer 1 Report
Comments and Suggestions for Authors
All requests were fully addressed.
Author Response
We would like to thank the reviewer for the thorough review and the positive assessment of our manuscript. We are pleased that all comments and suggestions were found to be fully addressed. We appreciate the reviewer’s time and valuable input.
Reviewer 2 Report
Comments and Suggestions for Authors
The authors have addressed all my comments
Author Response

(The authors gave the same response as above.)

Reviewer 5 Report
Comments and Suggestions for Authors
My comments and suggestions have been addressed
Author Response

(The authors gave the same response as above.)
